# Loneliness, Family Communication, and School Adjustment in a Sample of Cybervictimized Adolescents

**DOI:** 10.3390/ijerph17010335

**Published:** 2020-01-03

**Authors:** Elizabeth Cañas, Estefanía Estévez, Celeste León-Moreno, Gonzalo Musitu

**Affiliations:** 1Departament of Health Psychology, Miguel Hernández University, Avda. de la Universidad s/n, 03202 Alicante, Spain; eestevez@umh.es; 2Department of Education and Social Psychology, Pablo Olavide University, 41013 Seville, Spain; cmleomor@upo.es (C.L.-M.); gmusoch@upo.es (G.M.)

**Keywords:** loneliness, family communication, school adjustment, cyberbullying, victim, gender differences

## Abstract

The objective of this study was to compare individual, family, and social variables, such as the perception of loneliness, family communication, and school adjustment in a sample of 2399 Andalusian (Spanish) adolescents aged 12 to 18 (*M* = 14.63, *SD* = 1.91) suffering from cybervictimization (low, moderate, and high). The results show that adolescents suffering from high cybervictimization report more loneliness, more problematic communication with both parents, and worse school adjustment than the rest of the groups. Regarding gender, differences are observed in open communication with the mother and in the dimensions of school adjustment, being more favorable for girls. However, there were no significant differences between girls and boys in the loneliness variable. The interaction effects indicate, on the one hand, that female severe cybervictims present more avoidant communication with the mother than the other groups, and, on the other hand, that male cybervictims of all three groups and female severe cybervictims have lower academic competence than the group of female low cybervictims, followed by female moderate cybervictims. These data support the idea that, depending on its intensity and duration, cybervictimization affects girls and boys differently in terms of individual, family, and social variables.

## 1. Introduction

The ever-expanding use of information and communication technologies (ICTs) in developed countries is increasing the problem of cyberbullying among adolescents, especially in the last decade [1]. Cyberbullying has been defined as aggressive, repetitive, and deliberate peer-to-peer behavior, where a person or group uses ICTs to abuse a victim who cannot easily defend him- or her-self [2,3].

Drawing on the ecological approach [4], which suggests that human behavior is the result of a complex interaction between individuals and their broader environment, cyberbullying is seen as the result of complex transactions between individual, family, and social factors [5]. Thus, the ecological approach provides an organizational framework through which to identify the risk factors of this problem [6], to understand the potential impact of cybervictimization at the individual and environmental level, as well as to design interventions that include all the contexts linked to this issue [7].

Some studies show that cyberbullying should be studied from an ecological perspective. This statement is based on the nature of this problem, since cyberbullying has the potential to extend beyond victims, to bystanders not confined to the same class, grade, school, or country, emphasizing the importance of considering the influence of the environmental contexts of the adolescent [8,9]. This theory provides a useful framework for examining the interplay between environmental factors and adolescent cyberbullying. However, there is still little research applying this approach to the analysis of this problem. The interest of this research lies in the fact that it analyzes several variables relating to each of the contexts that distinguish the ecological approach, which are relevant to adolescents’ lives, to construct a deeper profile of cyberbullying victims, as yet an incipient issue in the investigation of cyberbullying. The main findings, from the reviewed literature, for each of the variables analyzed in this study, are shown below.

### Literature Review

On an individual level, previous research has shown that cybervictims report more feelings of loneliness than those not involved in cyberbullying [10]. The findings available so far suggest that feelings of loneliness can be a risk factor to become a cybervictim, but also a consequence of this role, especially in adolescents, where the use of social media is significant. For example, the situation of bullying can increase cybervictims’ feelings of loneliness [11], which can lead them to spend more hours using ICTs in order to reduce these feelings, thus providing more control and opportunities to communicate with others [12]. However, the use of online communication diminishes face-to-face social interactions and promotes weaker and more superficial social relationships [13], thus explaining the greater sense of loneliness and isolation reported by cybervictims [14]. 

In this sense, a positive family environment can favor cybervictims’ emotional state, even mitigating the feeling of loneliness they report [15]. However, scientific evidence has shown that cybervictims’ family relationships tend to be more problematic than those of adolescents not involved in cyberbullying, expressed in evasive and conflicting family communication [16]. This pattern of communication, in turn, makes it difficult for parents to transmit to their children the personal and social support and resources needed to deal with difficult situations (such as cyberbullying) [17,18,19].

Continuing the ecological contextualization, at the social level, cybervictimization can also spread to the school, adversely affecting the school adjustment of those involved. Some authors point out that cybervictims show greater social adjustment problems, such as fear of going to school, negative attitudes towards school, and poorer academic performance than the rest of students involved in cyberbullying [18,20]. In this sense, numerous scientific works underline the importance of family participation in the optimal development of children’s school adjustment [21,22]. Such family involvement refers to the parents’ involvement in their children’s educational lives and/or in activities or behaviors related to academic and social outcomes [23]. However, it is worth noting that victimization has been associated with parents’ low levels of involvement in their children’s education and schooling [24]. 

Another important factor for school adjustment of adolescents who are victimized is the quality of their interactions with their teachers. In this regard, Troop-Gordon and Kuntz [25] noted that close and supportive teacher–student relationships buffered the negative impact of bullying on the victims’ school adjustment. Despite this, most of the findings indicate that bullying victims report weaker and poorer relationships with teachers compared to adolescents not involved in bullying [26,27]. However, no information is available about the influence teachers may be having on adolescent students involved in the problem of cyberbullying. 

Although there is a solid knowledge base on the importance of individual, family, and social context in the study of cyberbullying, there are still gaps in the development of a cybervictim profile that takes into account each of these contexts. Thus, for example, the study of cyberbullying tends to leave out the school context, but this problem also negatively impacts on cybervictims’ school adjustment [18]. In this respect, the analysis of each dimension of school adjustment can provide a more complete picture of the roles associated with cyberaggression. In addition, there are generally few cyberbullying studies that analyze gender differences, even though the scientific literature notes that girls who suffer victimization tend to express more intense feelings of loneliness [28], greater difficulties to communicate with parents [29], and a poorer school adjustment [30] than boys. In addition, the studies do not comprehensively analyze the differentiating effect of communication with the father and the mother in cybervictims, which could be interesting to determine the role of communication of the two parents separately [31].

Taking these limitations into account, the objectives of this study were as follows: (1) In the individual and psychoemotional field, to analyze differences in the perception of the feeling of loneliness as a function of the degree of cybervictimization (low, moderate, and high); (2) in the family sphere, to analyze differences in family communication (open, offensive, and avoidant), with the father and mother separately, as a function of the degree of cybervictimization (low, moderate, and high); (3) at the school level, to analyze differences in school adjustment perceived by teachers, that is, in social adjustment, academic competence, family involvement in school, and quality of the teacher–student relationship, as a function of the degree of cybervictimization (low, moderate, and high); and finally, to analyze differences in the study variables (feelings of loneliness, family communication, and school adjustment) in both genders (boys and girls), and their interaction as a function of the degree of cybervictimization (low, moderate, and high). 

Based on the previous literature on cybervictimization in adolescence, the following hypotheses were established:
**Hypothesis 1** **(H1).***Victims exposed to incidents of more severe cyberbullying will show a greater sense of loneliness compared to those victims who suffer moderate or low cyberbullying*.
**Hypothesis 2** **(H2).***Victims exposed to incidents of more severe cyberbullying will show more problematic communication with both parents compared to those victims who suffer moderate or low cyberbullying*.
**Hypothesis 3** **(H3).***Victims exposed to incidents of more severe cyberbullying will show lower school adjustment in all its dimensions compared to those victims who suffer moderate or low cyberbullying*.
**Hypothesis 4** **(H4).***As for gender differences, girls will show greater feelings of loneliness, but better family communication and school adjustment than boys*.
**Hypothesis 5** **(H5).***Due to the interaction of gender with cyberbullying greater feelings of loneliness, more family communication problems, and poorer school adjustment will be observed in female cybervictims compared to male cybervictims*.

## 2. Materials and Methods

### 2.1. Ethics Approval

The Ethics Committee of the Pablo de Olavide University approved the study on 13 March 2015, as complying with the ethical values required for research with human beings and respecting the basic principles included in the Helsinki Declaration (Ref. PSI2012-33464).

### 2.2. Participants

Participant selection was done by means of stratified randomized cluster sampling. The sampling units were: the geographic area—rural (74%) and urban (26%)—as the primary unit, and the school—public (75%) and private/subsidized (25%)—as the secondary unit. The sample size—with a sampling error of ±2%, a confidence level of 95%, and *p* = *q* = 0.5—was estimated at 2399 students (50.2% boys and 49.8% girls), with ages between 12 to 18 years (*M* = 14.63, *SD* = 1.91), from 19 high schools from Andalusia (Spain), of which 12 are state-owned and 7 are private/subsidized. The distribution of the participants by gender and academic year was the following: 436 in the first year of compulsory secondary education (219 males and 217 females), 418 in the second year of compulsory secondary education (223 males and 195 females), 365 in the third year (183 males and 181 females), 388 in the fourth year (174 males and 214 females), 419 in the first year of Baccalaureate (203 males and 216 females), and 373 in second year of Baccalaureate (202 males and 171 females).

### 2.3. Instruments

UCLA Loneliness Scale [32], adapted to Spanish by Expósito et al. [33]. This scale consists of 20 items that provide a general measure of loneliness felt by the teenager (e.g., “I feel isolated from others” or “I can find companionship when I want it”). The responses range from 1 (never) to 4 (always). The scale presented a Cronbach alpha reliability in this study of 0.90.

The Parent–Adolescent Communication Scale (PACS) [34]; with Spanish adaptation by Musitu et al. [35] is composed of 20 items rated on a 5-point Likert-type scale ranging from 1 (never) to 5 (always). The items measure the adolescent’s perception of the communication with his/her father and mother separately. This scale has three subscales for the father and three for the mother: Openness in Father/Mother Communication (e.g., “I can discuss my beliefs with my mother/father without feeling restrained or embarrassed); Offensive Communication with Father/Mother (e.g., “S/he insults me when s/he is angry with me”); and Avoidant Communication with Father/Mother (e.g., “There are topics I avoid discussing with him/her”). A second-order confirmatory factor analysis (CFA) using the maximum likelihood estimation method confirmed the fit of the proposed measurement model, SBχ^2^(156) = 545.30, *p* < 0.001, CFI = 0.94, NNFI = 0.92, RMSEA = 0.04, 90% CI [0.000, 0.030]. The Cronbach’s alpha reliability coefficients in this study were: 0.91 for Open communication both with the mother and father; 0.70 and 0.69 for Offensive communication with the father and mother, respectively; and 0.65 and 0.67 for Avoidant communication with the father and mother, respectively. The Cronbach alpha reliability coefficient of this study for the full scale (communication with the mother and with the father) was 0.71.

The scale of Teacher’s Perception of School Adjustment (PROF-A) is an expanded and revised version of the “Escala de Evaluación del Alumno por el Profesor” (EA-P; Teacher’s Student Assessment Scale) of Cava and Musitu [36] based on an original idea of García-Bacete [37]. The PROF-A scale consists of 14 items and is an extension of the original scale. This new scale includes items relating to the teacher’s perception of the student’s degree of social adjustment (e.g., “student’s interest in knowing and befriending classmates”), the student’s academic performance and competence (e.g., “student’s participation in activities, discussions, debates, etc., proposed in class”), the family’s involvement (e.g., “degree of family involvement in the child’s homework”), and the teacher–student relationship (e.g., “the time I spend talking to this student”). Teachers rate each of the 14 items with a response ranging from 1 (very low) to 10 (very high), and complete a scale for each of their students. The Cronbach alpha reliability coefficient of this scale in this study was 0.93. Subscale reliabilities (Cronbach alpha) were 0.91 for Social adjustment, 0.94 for Academic performance and competence, 0.92 for Family involvement, and 0.85 for the Teacher’s relationship with the student.

Adolescent Victimization through Mobile Phone and Internet Scales (CYBVIC) was adopted from a peer victimization scale that has been widely used and validated in our context (see [38]), and the classification of Willard [39,40]. These scales were developed to measure bullying experienced through the mobile phone and the Internet. Items that best represented each category of Willard’s classification were selected and adapted to the situation of technological bullying. In cases where there were no representative items of the category, items were elaborated. Both scales (mobile phone and Internet) measure the bullying experienced over the past year with a response ranging from 1 (never) to 4 (always). The scale of victimization by mobile phone consists of 8 items that evaluate behaviors that involve incidents of harassment, persecution, denigration, violation of privacy, and social exclusion. The Cronbach alpha reliability coefficient in this study was 0.76. The scale of victimization over the Internet has the same response range as the previous scale and consists of the same 8 items, to which are added 2 more items related to aggressions of violation of privacy and impersonation. The Cronbach alpha reliability coefficient of this scale was 0.82.

### 2.4. Procedure

Data for this research were collected as part of a larger study on violent behavior in adolescents in Spain. First, a letter with a summary of the research project was sent to the selected schools as a first step. Subsequently, initial telephone contact with the school headmasters was established, followed by an informative seminar with all the teaching staff in each school, informing them of the objectives and methodology of the study during a two-hour presentation. In parallel, a letter describing the study was sent to the parents, requesting them to indicate in writing if they did not wish their child to participate in (1% of parents used this option). Passive consent was received by the rest of the parents.

Teachers and parents both expressed a wish to be informed about the main results of the investigation in a meeting with the research team. This meeting took place after data analyses were completed. The administration of the instruments was carried out by a group of trained and expert researchers in each region. Before data collection, students also attended a short briefing in which they provided written consent (none of the adolescents refused to participate). On the dates scheduled with the teaching staff, students were approached at their own classrooms in school to fill out the questionnaires voluntarily during a regular class period. The researchers of the present study administered the tools in the presence of the student’s tutor. The order of administration of the instruments was counterbalanced in each classroom and school. The instructions were read aloud, emphasizing the importance of answering all questions and the anonymity of the answers. During the administration of the tests, the researchers were present to resolve doubts and ensure an unbiased process. Regarding family communication, adolescents were asked to respond keeping in mind the person they perceived as their mother or father during the past year. If one parent was deceased, or the students do not have a relation with their father or mother, we did not consider the information. Students could refuse to answer if they found it difficult to do so. Their privacy was guaranteed, reducing any possible social desirability effects. The surveys that were suspicious in terms of the response patterns were not coded in the database (these surveys represented 1% of the total original samples).

### 2.5. Data Analyses

First, the cross-distribution of the adolescents was calculated according to the degree of cybervictimization and gender (see Table 1). Subsequently, a factorial multivariate analysis (MANOVA, 3 × 2) was performed with the SPSS statistical program (version 20, IBM Corp, Armonk, NY, USA), considering the clusters of cybervictimization (low, medium, and high) and gender (boys and girls) as fixed factors to analyze possible interaction effects. The variable loneliness, the three dimensions of family communication—open, offensive, and avoidant—and the four dimensions of the teacher’s evaluation—school adjustment, academic competence, family involvement, and teacher–student relationship—were considered dependent variables. Finally, univariate tests (ANOVAS) were calculated to study the differences in the statistically significant variables and the Bonferroni post-hoc test was performed (α = 0.05).

Analysis of mean differences based on the location of the school and its public or private status were not included in subsequent analyses with the target variables of the study as they were not statistically significant. Missing values were handled with the regression imputation method [41], and the outliers with the criteria provided by Hair et al. [42]. These values were those whose standardized scores had an absolute value greater than 4. For multivariate detection, the Mahalanobis distance was calculated. A multivariate outlier was considered if the probability associated with the Mahalanobis distance was 0.001 or less [43].

## 3. Results

### 3.1. Adolescents’ Distribution and Interaction Effects of Cybervictimization and Gender

When analyzing the adolescents’ distribution according to the degree of cybervictimization and gender (see Table 1), the highest frequency was observed in low cybervictims (43%), followed by moderate (33.6%) and severe (23.4%) cybervictims. Regarding gender, the prevalence in girls of moderate (37.3%) and severe (23.8%) cybervictims was higher than that of the boys (29.9% and 22.9%, respectively). Conversely, regarding low cybervictimization, the prevalence in boys was higher (47.2%) than that in girls (38.8%). 

Statistically significant differences were found in the main effects of gender (Λ = 0.944, F(11, 2383) = 12.796, *p* < 0.001, *ηp*^2^ = 0.056) and cybervictimization (Λ = 0.891, F(22, 4776) = 12.893, *p* < 0.001, *ηp*^2^ = 0.056. Also, two statistically significant interaction effects were obtained between gender and cybervictimization (Λ = 0.984, F(22, 4776) = 1,735, *p* < 0.05, *ηp*^2^ = 0.008) (see Table 2). 

### 3.2. Cybervictimization and Target Variables of the Study

Regarding cybervictimization, the results of the ANOVA (see Table 3) showed significant differences between groups in the variables of loneliness, F(2, 2396) = 70.73, *p* < 0.001, Open communication with mother, F(2, 2396) = 22.42, *p* < 0.001, Offensive communication with mother, F(2, 2396) = 45.25, *p* < 0.001, Avoidant communication with mother, F(2, 2396) = 18.03, *p* < 0.001, Open communication with father, F(2, 2396) = 47.72, *p* < 0.001, Offensive communication with father, F(2, 2396) = 44.42, *p* < 0.001, Avoidant communication with father, F(2, 2396) = 14.19, *p* < 0.001, Social adjustment, F(2, 2396) = 7.63, *p* < 0.001, Academic competence, F(2, 2396) = 31.60, *p* < 0.001, Family involvement, F(2, 2396) = 14.14, *p* < 0.001, and relationship of teacher with the student, F(2, 2396) = 11.94, *p* < 0.001. The Bonferroni tests (α = 0.05) indicated that severe cybervictims scored higher on the variables loneliness and offensive communication with mother and father than did moderate and low cybervictims, respectively. In turn, severe cybervictims scored lower on the variables open communication with mother and father than did moderate and low cybervictims, respectively. In addition, severe and moderate cybervictims scored higher in avoidant communication with mother and father compared to low cybervictims. Finally, moderate and low cybervictims scored higher on all the dimensions of school adjustment—social adjustment, academic competence, family involvement, and positive relationship with the teacher—than did severe cybervictims.

### 3.3. Gender and Target Variables of the Study

As noted in Table 4, girls obtained higher scores than boys in Open communication with the mother, F(1, 2397) = 11.71, *p* < 0.01, Social adjustment, F(1, 2397) = 36.17, *p* < 0.001, Academic competence, F(1, 2397) = 74.81, *p* < 0.001, Family involvement, F(1, 2397) = 4.77, *p* < 0.05, and Teacher–student relationship, F(1, 2397) = 25.54, *p* < 0.001. On the other hand, no significant gender effects were found in loneliness, offensive communication with mother and father, in avoidant communication with mother and father, and open communication with father. 

### 3.4. Cybervictimization, Gender and Target Variables of the Study

Two statistically significant interaction effects were obtained between cybervictimization and gender (see Table 5), in the variable avoidant communication with the mother, F(2, 2393) = 3.174, *p* < 0.05, and in academic competence, F(2, 2393) = 3.082, *p* < 0.05 (see Figure 1). 

Female severe cybervictims obtained the highest scores in avoidant communication with the mother, with differences that are highly significant, both in relation to the girls themselves and to the male low cybervictims. In addition, scores on avoidant communication with the mother were higher in male moderate and severe cybervictims and in female moderate cybervictims compared to those of female low cybervictims. Male cybervictims (in all three groups) and female severe cybervictims scored significantly lower in academic competence than did female low cybervictims, followed by female moderate cybervictims. 

## 4. Discussion

The main objective of this work was to analyze differences in perception of loneliness, family communication, and school adjustment in adolescent boys and girls depending on the intensity and duration of cybervictimization (low, moderate, and high). Cybervictimization and gender were also analyzed independently, in order to determine the specific contribution of the two variables to each of the target variables. 

The first hypothesis proposed in this study postulated that severe cybervictims would have a greater sense of loneliness compared to moderate and low cybervictims. Consistent with this hypothesis, the young people in this study who suffered from cybervictimization with greater intensity and duration reported greater feelings of loneliness than the rest of the cybervictims. Previous studies, such as that of Larrañaga et al. [14], indicate that cybervictims report feeling lonelier and marginalized by the peer group and identify fewer friends compared to non-victimized adolescents. However, victims of cyberbullying sometimes try to escape the feeling of loneliness through the use of ICTs, ignoring that the time invested in the network and technological devices can aggravate the state of isolation from which they are trying to escape [44].

The second hypothesis of this study suggested that severe cybervictims would show more problematic communication with both parents. The results confirm this hypothesis, as severe cybervictims reported maintaining less open, more offensive, and more avoidant communication both with the father and the mother than low and moderate cybervictims. These results coincide with those obtained by other authors who emphasize that severe cybervictims have more problematic parent–child communication than other groups of cybervictims [14]. Recent research suggests that family communication problems may be due to the fact that cybervictims avoid sharing the experience of cyberbullying with their parents because they do not think that their parents can put an end to the problem and they fear that this will lead to further reprisals by their peers [16]. 

As cybervictimization can occur among schoolmates, the third hypothesis postulated that severe cybervictims would present greater problems in each of the dimensions of school adjustment than other cybervictims. The results confirm this hypothesis, as the severe cybervictims in this study reported worse social adjustment, less academic competence and family involvement in school, as well as poorer relationships with the teachers. Regarding social adjustment, previous studies have indicated that cybervictims show more shyness, vulnerability, insecurity towards classmates, and social avoidance than adolescents not involved in cyberbullying [45]. 

Severe cybervictims also showed less academic competence than the other cybervictims in this study. This result coincides with research such as that of Wright [46], who, in addition to finding lower grades and higher truancy in cybervictims, explained this finding through the negative impact of cyberbullying on the motivation and participation of the victims.

As for the lower family involvement in the school shown by the severe cybervictims of this study, there are no previous studies analyzing the differences in parental involvement in the school based on the degree of cybervictimization. However, previous research indicates that the lack of parents’ involvement in school activities can act as a predictor of victimization among young people [47], which seems to be confirmed by the cybervictims of the present work. 

Regarding the poorer relationships that the severe cybervictims of this study have with their teachers, this is in line with the results of previous research showing that victims of bullying have more conflictive and cooler relationships with the teachers [27,48]. 

It is clear that cyberbullying plays an important role in the target variables, but so does gender, as in the fourth hypothesis of this study, which suggested that girls would show greater loneliness but better family communication and school adjustment than boys. The results obtained allow us to partially confirm the hypothesis, because, despite the fact that the girls in this study had more open communication with their mothers and better school adjustment in general—which implies better social adjustment, greater academic competence and family involvement—as well as a better relationship with the teacher than the boys, there were no differences in terms of loneliness or in the other dimensions of parent–child communication. 

The results related to communication are in line with previous research indicating that girls tend to communicate more openly with their mothers than boys, and suggesting that same-gender membership promotes this communication between them because, at this stage of development, daughters perceive their mothers as learning models and supportive figures with whom to share confidence [49]. Besides, and according to this result, adolescent boys are less dependent on parents than girls, so that boys tend to communicate to a lesser extent with them [50]. 

The increased social adjustment and academic competence shown by the girls in this study also coincide with previous studies indicating that girls tend to have better overall academic achievement than boys [51]. The greater importance that girls attach to maintaining warm and supportive social relationships with their peer group [52], and their higher commitment and concern about their academic success [53], may explain the greater social adjustment and academic competence of the girls of this study compared to the boys.

Continuing with the results regarding school adjustment, higher family involvement in the girls’ educational experiences compared to that of the boys converges with previous research [54]. Thus, in line with the above about the increased concern that girls experience compared to boys about academic success, some authors, such as Patrick et al. [55], suggest that this concern may motivate them to resort to their parents more often than boys to ask them for help with their homework, thus promoting more parental commitment towards the school life of their daughters. The data also indicate that girls have warmer relationships with their teachers than do boys. This result is in line with the findings of Glüer and Gregoriadis [56], who also indicate girls’ greater willingness to interact socially explains their being more receptive to teachers and to maintaining closer relationships with them.

In relation to the fifth hypothesis, it was suggested that significant interactions would be observed in which female cybervictims would show more feelings of loneliness, more family communication problems, and worse school adjustment than male cybervictims. Although significant interactions were obtained, this hypothesis is ruled out because only female severe cybervictims showed more avoidant communication with the mother than did male low cybervictims, and less academic competence than did female low and moderate cybervictims, although this was also observed in all three groups of male cybervictims. 

In the case of avoidant communication with the mother, although some research indicates that female severe cybervictims tend to communicate more evasively with the maternal figure [14], there are no previous studies that have thoroughly analyzed the communication between mother and daughter with regard to cyberbullying. However, the fact that girls tend to be more sensitive to the negative reactions of their parents [57] and to avoid the issue of bullying with them [18] may explain the gender differences observed in terms of the avoidant communication of the female cybervictims of this study.

Although to date there are no data on the differences between male and female cybervictims in academic competence, the higher academic competence observed in this study of female low cybervictims followed by female moderate cybervictims may have an explanation. In accordance with the above results, the warmer relationships with teachers observed in moderate and low cybervictims, and, as a function of gender, in girls, may buffer the negative effects of victimization on the school adjustment of these groups of cybervictims [25]. This, in turn, could justify the lower academic competence of the male cybervictims in all three groups, and of the female severe cybervictims of this study.

### Limitations of the Study

Despite the novel contribution of this work, it also presents some limitations which the authors acknowledge should be remedied in future research. On the one hand, the results obtained derive only from the role of cybervictim, and they cannot be generalized to students of other educational levels, as in the different courses of primary education where cyberaggression problems are also observed. On the other hand, the results presented could have biases derived from social desirability in the responses issued in the self-reported scales, although previous studies have found acceptable reliability and validity of such instruments for measuring risky behaviors in adolescents [18]. Besides, regarding the administration of the instruments, no previous screening of the participants was performed, so it could be possible that in some cases victims and bullies were in the same room during the data collection. Finally, the cross-sectional nature of the data prevents the establishment of causal inferences between the variables. 

It would be interesting that future work includes samples of students from different academic levels. In addition, it would be recommendable to add several instruments to evaluate the adolescent’s behavior (by peers, teachers, and parents), since that could help to analyze the inter-source concordance. Also, victims and bullies should be separated during the data collection. Future research should also perform longitudinal studies that help to confirm the directions of influence between variables. Finally, it would be interesting to deepen the understanding of gender differences related to cybervictimization.

## 5. Conclusions

In conclusion, the results of this study expand our knowledge about the psychosocial and family profile of cybervictims, and support the idea that cybervictimization has differential effects on boys and girls. The findings generally suggest that, as the intensity and duration of cybervictimization increase, the feelings of loneliness and problems of family communication and school adjustment in cybervictims increase. And in particular, girls tend to avoid communication with the maternal figure, as well as to reduce their academic competence, a result that is also seen in the male cybervictims of all three groups. Accordingly, this study highlights the importance of the ecological approach to the study and comprehension of cyberbullying and, therefore, for the design of interventions targeting this problem. 

## Figures and Tables

**Figure 1 ijerph-17-00335-f001:**
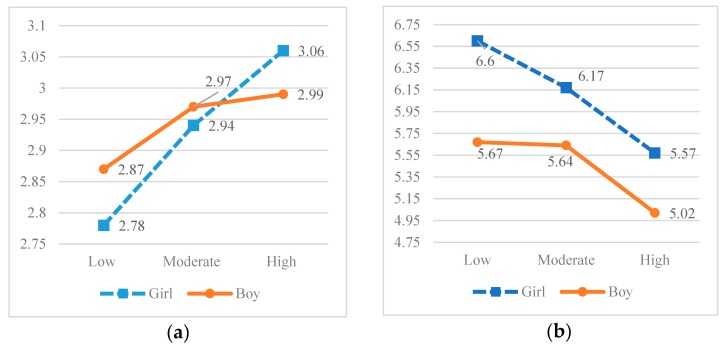
(**a**) Interaction between avoidant communication with the mother, degree of cybervictimization and gender; (**b**) Interaction between academic competence, degree of cybervictimization and gender.

**Table 1 ijerph-17-00335-t001:** Distribution of adolescents according to cybervictimization and gender clusters.

Gender	Total Sample *N* (%)	Victimization		
		Low (*n* = 1032) *N* (%)	Moderate (*n* = 806) *N* (%)	High (*n* = 561) *N* (%)
Boys	1204 (50.2%)	568 (47.2%)	360 (29.9%)	276 (22.9%)
Girls	1195 (49.8%)	464 (38.8%)	446 (37.3%)	285 (23.8%)
Total	2399 (100%)	1032 (43.0%)	806 (33.6%)	561 (23.4%)

**Table 2 ijerph-17-00335-t002:** Factorial MANOVA (4 × 2).

	Λ	*F*	*df_between_*	*df_error_*	*p*	*η* ^2^
(A) Cybervictimization	0.891	12.893	22	4776	<0.001 ***	0.056
(B) Gender	0.944	12.796	11	2383	<0.001 ***	0.056
A × B	0.984	1.735	22	4776	<0.05 *	0.008

Note: Λ: wilks lambda; F: contrast statistic of MANOVA; * *p* < 0.05. *** *p* < 0.001.

**Table 3 ijerph-17-00335-t003:** Means, standard deviations (SD), and ANOVA results between cybervictimization, loneliness, family communication, and school adjustment.

	Cybervictimization				
	Low	Moderate	High	*F*(2, 2396)	*η* ^2^ *_p_*
Loneliness	1.77 (0.42) ^c^	1.85 (0.38) ^b^	2.03 (0.47) ^a^	70.73 ***	0.056
Open communication mother	3.96 (0.84) ^a^	3.86 (0.71) ^b^	3.68 (0.83) ^c^	22.42 ***	0.018
Offensive communication mother	1.65 (0.70) ^c^	1.79 (0.68) ^b^	2.00 (0.75) ^a^	45.25 ***	0.036
Avoidant communication mother	2.83 (0.71) ^b^	2.95 (0.64) ^a^	3.03 (0.67) ^a^	18.03 ***	0.015
Open communication father	3.72 (0.86) ^a^	3.54 (0.74) ^b^	3.30 (0.85) ^c^	47.72 ***	0.038
Offensive communication father	1.64 (0.67) ^c^	1.78 (0.67) ^b^	1.98 (0.75) ^a^	44.42 ***	0.036
Avoidant communication father	2.90 (0.73) ^b^	3.05 (0.64) ^a^	3.06 (0.68) ^a^	14.19 ***	0.012
Social adaptation	6.93 (1.40) ^a^	6.94 (1.39) ^a^	6.66 (1.59) ^b^	7.63 ***	0.006
Academic competence	6.09 (1.86) ^a^	5.93 (1.94) ^a^	5.30 (2.07) ^b^	31.60 ***	0.026
Family involvement	6.41 (1.79) ^a^	6.21 (1.81) ^a^	5.90 (1.99) ^b^	14.14 ***	0.012
Teacher’s relationship with the student	7.26 (1.19) ^a^	7.19 (1.23) ^a^	6.94 (1.44) ^b^	11.94 ***	0.010

Note: F = Fisher–Snedecor’s F; ^a^ = boys low cybervictims; ^b^ = boys moderate cybervictims; ^c^ = boys high cybervictims; Bonferroni Test = ^a^ > ^b^ > ^c^; *η*^2^*_p_* = partial square eta. The data in parentheses correspond to the standard deviations. *** *p* < 0.001.

**Table 4 ijerph-17-00335-t004:** Means, standard deviations (SD), and ANOVA results between gender, loneliness, family communication, and school adjustment.

	Gender			
	Girl	Boy	*F*(1, 2397)	*η* ^2^ *_p_*
Loneliness	1.86 (0.45)	1.85 (0.41)	0.69	0.000
Open communication mother	3.92 (0.79)	3.80 (0.82)	11.71 **	0.005
Offensive communication mother	1.79 (0.71)	1.77 (0.72)	0.58	0.000
Avoidant communication mother	2.91 (0.68)	2.93 (0.68)	0.51	0.000
Open communication father	3.53 (0.83)	3.59 (0.85)	2.43	0.001
Offensive communication father	1.77 (0.69)	1.76 (0.71)	0.14	0.000
Avoidant communication father	3.01 (0.68)	2.97 (0.70)	2.48	0.001
Social adaptation	7.05 (1.39)	6.69 (1.48)	36.17 ***	0.015
Academic competence	6.19 (1.94)	5.51 (1.92)	74.81 ***	0.030
Family involvement	6.31 (1.84)	6.14 (1.87)	4.77 *	0.002
Teacher’s relationship with the student	7.30 (1.28)	7.04 (1.25)	25.54 ***	0.011

Note: F = Fisher–Snedecor’s F; *η*^2^*_p_* = partial eta squared. The data in parentheses correspond to the standard deviations. * *p* < 0.05. ** *p* < 0.01. *** *p* < 0.001.

**Table 5 ijerph-17-00335-t005:** Means, standard deviations (SD), and ANOVA results between cybervictimization, gender, avoidant communication with mother, and academic competence.

	Gender	Cybervictimization					
		Low	Moderate	High	*F*(2, 2393)	*η* ^2^ *_p_*	Post Hoc
ACM	Boy	2.87 (0.72) ^a^	2.97 (0.62) ^b^	2.99 (0.68) ^c^	3.174 ***	0.017	^f^ > ^a, d^
	Girl	2.78 (0.69) ^d^	2.94 (0.65) ^e^	3.06 (0.66) ^f^			^b, c, e^ > ^d^
AC	Boy	5.67 (1.82) ^a^	5.64 (1.90) ^b^	5.02 (2.06) ^c^	3.082 ***	0.060	^d^ > ^a, b, c, e, f^
	Girl	6.60 (1.77) ^d^	6.17 (1.94) ^e^	5.57 (2.04) ^f^			^e^ > ^a, b, c, f^

Note: ACM = avoidant communication mother; AC = academic competence; ^a^ = boys low cybervictims; ^b^ = boys moderate cybervictims; ^c^ = boys high cybervictims; ^d^ = girls low cybervictims; ^e^ = girls moderate cybervictims; ^f^ = girls high cybervictims; F = Fisher–Snedecor’s F; Bonferroni test α = 0.05; *η*^2^*_p_* = partial eta squared. The data in parentheses correspond to the standard deviations. *** *p* < 0.001.

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
