# Peer review of "Loneliness, Family Communication, and School Adjustment in a Sample of Cybervictimized Adolescents"

_ijerph, 2020, doi:10.3390/ijerph17010335_

Round 1
Reviewer 1 Report
Thank you for the opportunity to review this very interesting manuscript entitled "Loneliness, Family Communication and School Adjustment in a Cybervictimized Adolescents Sample". The authors have done an effort to analyze the influence of the variables with the ecological perspective on cybervictims. I would like to congratulate the authors for this manuscript, I consider that this research should be published in this journal. I consider minor changes should be done for this manuscript. For this reason, I recommend an improvement of the manuscript with these suggestions.
Title:
I consider that the title “Loneliness, Family Communication and School Adjustment in a Cybervictimized Adolescents Sample” show the meaning of the manuscript. The title of this manuscript is concise and specific.
Abstract:
The abstract should be a total of about 200 words maximum. This manuscript have a total of 215 words. For this reason, I recommend the authors reduce some words of the abstract according the instructions for authors of this journal. Also, I consider, that the abstract should be show the results of the differences between girls and boys in the loneliness variable.
https://www.mdpi.com/journal/ijerph/instructions
Introduction:
I recommend that the authors describe some previous studies that analyses the cyberbullying according this perspective in the second paragraph.
Method:
Participants
In this section, the authors should describe in the participants section the distribution of the adolescents according to the specific grade. In the procedure, the authors should add in the second paragraph other options that justify the not answer of the adolescents in this instrument.
“Regarding family communication, adolescents were asked to respond keeping in mind the person they perceived as their mother or father during the past year. If one parent was deceased, we did not consider the information. Students could refuse to answer if they found it difficult to do so.”
The authors could add: If one parent was deceased, or it does not have a relation with him father or her mother, we did not consider the information.
Results:
The authors have done a great job in this section. They provide a precise and clear description of the results, and the reader can understand the results very easy. I recommend to change one aspect of the figure 1. In the section b) the number is hidden by the line.
Discussion and conclusions:
The authors should describe more an explanation about the result of less open communication with mother in boys. Also, the authors should include the future lines of research according these results, probably they can include this in the final of this section.
References:
The style of the references is correct. Although the authors should include the changes that they realized as a result of the revisions in this section.
Author Response
Title:
I consider that the title “Loneliness, Family Communication and School Adjustment in a Cybervictimized Adolescents Sample” show the meaning of the manuscript. The title of this manuscript is concise and specific.
Abstract:
The abstract should be a total of about 200 words maximum. This manuscript have a total of 215 words. For this reason, I recommend the authors reduce some words of the abstract according the instructions for authors of this journal. Also, I consider, that the abstract should be show the results of the differences between girls and boys in the loneliness variable.
The abstract has been reduced to 192 words and shows the result of gender differences in the loneliness variable.
Introduction:
I recommend that the authors describe some previous studies that analyses the cyberbullying according this perspective in the second paragraph.
Some previous studies has been added to introduce second paragraph.
Method:
Participants
In this section, the authors should describe in the participants section the distribution of the adolescents according to the specific grade. In the procedure, the authors should add in the second paragraph other options that justify the not answer of the adolescents in this instrument.
“Regarding family communication, adolescents were asked to respond keeping in mind the person they perceived as their mother or father during the past year. If one parent was deceased, we did not consider the information. Students could refuse to answer if they found it difficult to do so.”
The authors could add: If one parent was deceased, or it does not have a relation with him father or her mother, we did not consider the information.
The education grade of participants is now described and it has been added the sentence suggested by the reviewer.
Results:
The authors have done a great job in this section. They provide a precise and clear description of the results, and the reader can understand the results very easy. I recommend to change one aspect of the figure 1. In the section b) the number is hidden by the line.
The number of this graphic is visible now.
Discussion and conclusions:
The authors should describe more an explanation about the result of less open communication with mother in boys. Also, the authors should include the future lines of research according these results, probably they can include this in the final of this section.
It has been explained the result about the less open communication with mother in boys and it has been included a section for future lines of research according to the results of the study.
References:
The style of the references is correct. Although the authors should include the changes that they realized as a result of the revisions in this section.
New references have been added according to new citations included in introduction section.

Reviewer 2 Report
Thank you for the opportunity to read this quite fascinating paper. The methods, results and implications are good. I have a few comments for some minor changes or revisions.
The term "sex"is used throughout - this is shifting in English with more of a gender v sex terminology. Some might argue that we should now be offering more choices than male or female but none the less, it is better if the shift to gender is done throughout the paper. on p. 2 Line 47 - sentence beginning the following - it feels like the sentence needs clarity to ensure the reader understands this means findings from the literature or research reviewed. I also wonder about a sub heading Literature Review. I would move the ethics approval up front in section 2. p. 4 line 179. There should be a period after team and then start a new sentence. In procedure, (or maybe in limitations) some thought should be given that it is quite probable that the victim and the bully were in the same room when data was being gathered. In procedure, please clarify who actually administered the tools, what instructions were given, how tools collected and how the room was managed during data collection. I know there is a general description of this, but it seems more detail will help the reader to understand how these issues were managed in the hope of diminishing data collection noise. A sub heading for Limitations on around line 370 would be appreciated. Author contributions - there are a couple of boiler place lines from the template that are instructional and not meant to be kept.
Author Response
The term "sex" is used throughout - this is shifting in English with more of a gender v sex terminology. Some might argue that we should now be offering more choices than male or female but none the less, it is better if the shift to gender is done throughout the paper.
The term "sex" has been changed by "gender" throughout the manuscript.
On p. 2 Line 47 - sentence beginning the following - it feels like the sentence needs clarity to ensure the reader understands this means findings from the literature or research reviewed. I also wonder about a subheading Literature Review.
The sentence has been clarified, and moreover, the subheading Literature Review has been added.
I would move the ethics approval up front in section 2.
Ethics approval has been moved to the beginning of section 2.
4 line 179. There should be a period after team and then start a new sentence.
It has been added a period after team, continuing with a new sentence.
In procedure, (or maybe in limitations) some thought should be given that it is quite probable that the victim and the bully were in the same room when data was being gathered.
This suggestion has been included in the limitations section.
In procedure, please clarify who actually administered the tools, what instructions were given, how tools collected and how the room was managed during data collection. I know there is a general description of this, but it seems more detail will help the reader to understand how these issues were managed in the hope of diminishing data collection noise.
It has been clarified the procedure section, explaining relevant details about this issue.
A subheading for Limitations on around line 370 would be appreciated.
This subheading has been added.
Author contributions - there are a couple of boiler place lines from the template that are instructional and not meant to be kept.
The instructional lines from the template have been removed.
Reviewer 3 Report
My comments are:
1) in the abstract it appears M = 63 and in section 2.1 M = 14.63, which I think is the correct value.
2) it is suggested to present each hypothesis separately.
3) It is suggested to include the sociodemographic characteristics of the sample and the distribution of students by academic level.
4) it is suggested to present the limitations in a separate section.
5) it is suggested to add future work.
Author Response
1) in the abstract it appears M = 63 and in section 2.1 M = 14.63, which I think is the correct value.
The value has been corrected.
2) it is suggested to present each hypothesis separately.
Each hypothesis is presented separately now.
3) It is suggested to include the sociodemographic characteristics of the sample and the distribution of students by academic level.
Sample characteristics have been described in more detail, as well as the distribution of students by academic level.
4) it is suggested to present the limitations in a separate section.
Limitations have been presented in a separate section at the end of the discussion.
5) it is suggested to add future work.
It has been included a section for future lines of research according to the results of the study.

Reviewer 4 Report
It seemed the study was done to prove the results were consistent with earlier studies on the same subject. The results of the first four hypotheses were were consistent with earlier studies. The fifth hypothesis was ruled out. It was suggested the warmer relationships with teachers observed in girls could lower academic competence of the male cyber victims. The conclusion has is that the idea of cybervictimization has differential effects on boys and girls. It seems most research would find this same conclusion. I did not find any shattering ideas claiming research on boys and girls would end with similar results.
Author Response
It seemed the study was done to prove the results were consistent with earlier studies on the same subject. The results of the first four hypotheses were were consistent with earlier studies. The fifth hypothesis was ruled out. It was suggested the warmer relationships with teachers observed in girls could lower academic competence of the male cyber victims. The conclusion has is that the idea of cybervictimization has differential effects on boys and girls. It seems most research would find this same conclusion. I did not find any shattering ideas claiming research on boys and girls would end with similar results.